# The Associations between Daytime Physical Activity, While-in-Bed Smartphone Use, Sleep Delay, and Sleep Quality: A 24-h Investigation among Chinese College Students

**DOI:** 10.3390/ijerph19159693

**Published:** 2022-08-06

**Authors:** Hongxing Chen, Guodong Zhang, Zhenhuan Wang, Siyuan Feng, Hansen Li

**Affiliations:** 1Department of Physical Education, Nanjing Institute of Technology, Nanjing 211167, China; 2Institute of Sports Science, College of Physical Education, Southwest University, Chongqing 400715, China; 3Institute for Health and Sport (iHeS), Victoria University, Melbourne 3011, Australia; 4Laboratory of Genetics, University of Wisconsin-Madison, Madison, WI 53706, USA

**Keywords:** smartphone, internet addiction, young adult, electronic screen, problematic behavior

## Abstract

Bedtime smartphone use is an emerging issue that threatens the sleep health of children and young adults. Physical activity can have numerous health benefits, including reducing problematic or addictive behavior. However, the role of daily physical activity in reducing bedtime smartphone use is understudied. Hence, we conducted a one-day cross-sectional on the weekend (21–22 May 2021) to investigate the associations between daytime physical activity, bedtime smartphone use, and sleep quality. A total of 828 college students were recruited in two colleges. Their daytime physical activity indices were captured, including self-reported physical activity duration, intensity, volume, and smartphone-monitored walking steps. The participants reported whether they used smartphone while lying in bed (before sleep) and whether they delayed sleep due to smartphone use. Their while-in-bed screen time (duration) and subsequent sleep quality were also measured with self-report and a numeric rating scale, respectively. The results suggested that daytime physical activity duration was associated with lower chances of while-in-bed smartphone use (OR = 0.907, *p* = 0.019) and smartphone-related sleep delay (OR = 0.932, *p* = 0.014). However, no significant association was found between physical activity indices and while-in-bed screen time or sleep quality. These findings may contribute to understanding the reciprocal relationship between physical activity and smartphone use and highlighting the potential of controlling problematic bedtime smartphone use through daily physical activity. Future research is warranted to examine the associations with extra objective measures.

## 1. Introduction

With the development of internet technology and electronic equipment, accessing the Internet is becoming part of everyday life, and screen time has naturally increased globally over the past few decades [1]. China is no exception. The latest national report has revealed that the number of Chinese Internet users has reached 1.1 billion, which exceeds the total population of many countries [2]. Among those Internet users, 99.6% mainly access networks via smartphone [2], indicating that the smartphone is becoming the most critical device that contributes to the increasing screen time. The popularity of smartphone elicits a range of health issues, including increased risks of myopia [3], poor mental health [4], and neck disability [5]. Smartphone-related sleep issues have attracted plenty of attention. Smartphone overuse may reduce sleep duration and quality [6,7], which have threatened the development of adolescents [8].

### 1.1. The Role of While-in-Bed Smartphone Use on Sleep Delay and Sleep Quality

Nighttime smartphone use is a rising research topic as it threatens sleep quality [9]. Smartphone screen lights may reduce melatonin and therefore reduce sleep quality [10]. Specifically, humans’ circadian rhythm is regulated by the suprachiasmatic nucleus, which uses external cues, such as light, to affect melatonin secretion and eventually control sleep-wake cycling [11]. Among the various spectrums, short wavelengths, usually perceived as blue light [12], are widely generated by LED screens [13] and are believed to have the most substantial impact on the suprachiasmatic nucleus [14]. Further, automatic smartphone use before sleep may be another critical reason to decrease sleep quality because losing track of time is common in smartphone use that can cause sleep delay [15]. Currently, numerous studies have suggested that nighttime smartphone use can lead to bedtime procrastination [16,17,18], which refers to “going to bed later than intended, without having external reasons for doing so” [19,20]. However, bedtime procrastination may not necessarily indicate sleep delay because going to bed is not simply equal to sleeping [21]. Compared to traditional electronic devices, smartphones are more convenient when lying in bed [21]. Therefore, people may watch TikTok and YouTube while lying in bed, which means that people may go to bed on time but fail to sleep on time due to smartphone use [21]. Such behavior is defined as a new concept called “While-in-Bed Procrastination,” which is regarded as a cause of sleep delay and decreased sleep duration [21]. Moreover, since more people tend to use their smartphones in the dark when lying in bed [22], their smartphone use can be particularly harmful to sleep quality [8]. Therefore, there is a demand to find factors likely to reduce while-in-bed smartphone use.

### 1.2. The Role of Physical Activity on While-in-Bed Smartphone Use and Sleep Quality

Physical activity (PA) is negatively associated with smartphone use [23,24]. Although the causal relationship is not confirmed yet, some scholars have assumed that physically active individuals may be less likely to be problematic Internet users [25]. Theoretically, PA is positively associated with self-regulation [26]. Meanwhile, self-regulation is negatively associated with procrastination [27], including bedtime procrastination [16,20]. These clues imply the potential role of PA in reducing smartphone use.

Moreover, PA is a protective factor for sleep quality [28]. Despite some controversies, many controlled trials have proved that PA may enhance sleep quality [29]. Therefore, it is reasonable to expect positive effects of daytime PA on smartphone users’ sleep quality. To date, the associations between daytime PA and smartphone-related sleep issues are understudied. Hence, this study aimed to examine the association between daytime PA, while-in-bed smartphone use, sleep delay, and sleep quality. To ensure temporal predominance, we conducted a questionnaire survey under a 24 h timeframe. Our hypotheses were:(1)Daytime PA is negatively associated with the chance and duration of while-in-bed smartphone use;(2)Daytime PA is negatively associated with the chance of smartphone-related sleep delay;(3)Daytime PA is positively associated with sleep quality.

## 2. Materials and Methods

### 2.1. Participants

We conducted a questionnaire survey via the “Wechat” platform during 21–22 May (a weekend), 2021. We introduced the topic of study as “An investigation on smartphone usage and health.” The recruiting information was delivered to 50 classes in two colleges. Our inclusion criteria were (1) college students at school and (2) smartphone users. Our exclusion criteria were (1) under quarantine and (2) having disabilities that prevent normal PA. Those who did not sign the informed consent or did not finish the questionnaire were not included. Eventually, a total of 828 participants volunteered to participate and completed the questionnaire.

### 2.2. Procedure

The investigation was open for participation in the morning. To ensure temporal precedence, we deployed questions in a time sequence. Specifically, we required the respondents to recall their daytime PA conditions, while-in-bed smartphone use, while-in-bed screen time, and overall sleep quality in a time sequence (Figure 1). Participants were required to participate using a WeChat account linking to their personal legal identity, and their device and IP address were monitored so that every person could only participate once.

### 2.3. Measurements

#### 2.3.1. Physical Activity

We deployed four items for measuring PA.

The first was the self-reported PA duration. The question was as follows: “How long did you generally perform PA (e.g., walking, biking, running, and other exercise or sports) during yesterday’s daytime?” Responses were captured by a 10-point Likert-type scale (from 0 to over 90 min with intercepts of 10 min).

The second was the self-reported PA intensity. The question was as follows: “How about the general intensity of the PA you performed yesterday’s daytime?” Responses were captured by a 10-point Likert-type scale (0 = very easy (tiny changes in breathing and heartbeat than resting) and 10 = extremely intensive (breathing and heartbeat are very fast)).

The third was the PA volume. Empirically, the product of duration, intensity, and frequency may indicate the total PA volume [30]. However, this method is inappropriate for a one-day study design, so we used the product of PA duration and intensity to form an extra variable to indicate the PA volume.

The fourth was the walking step, which is detected by the gyroscope, gravity sensor, and step counting intelligent module in smartphones. The walking steps were evaluated by the “Wechat sports,” a smartphone application bonded to participants’ Wechat accounts. We asked participants to read their records and report if this function was available.

#### 2.3.2. While-in-Bed Smartphone Use

We deployed two items for investigating smartphone use.

The first was the chance of while-in-bed smartphone use. The question was as follows: “Did you use smartphone before sleep (when you were lying in bed) yesterday?” The responses were coded as 0 = No and 1 = yes.

The second item was while-in-bed screen time, which referred to the duration of while-in-bed smartphone use. The question was as follows: “How long did you use smartphone before sleep (when you were lying in bed) yesterday night?” The responses were captured by a 10-point Likert-type scale (from 0 to over 90 min with intercepts of 10 min).

#### 2.3.3. Smartphone-Related Sleep Delay

The question for smartphone-related sleep delay was: “Did you sleep later than you intended, planned, or wanted due to smartphone use yesterday night?” The responses were coded as: 0 = No and 1 = yes.

#### 2.3.4. Sleep Quality

Participants were required to indicate their overall sleep quality with a 11-point numeric rating scale ranging from 0 (“best possible sleep”) to 10 (“worst possible sleep”) [31].

#### 2.3.5. Sociodemographic Characteristics

The gender of the respondents was categorized into two levels (1 = male and 2 = female.). Ages were captured with an open-ended box. Family monthly incomes were categorized into seven levels (RMB 0–6000, RMB 6001–10,000, RMB 10,001–14,000, RMB 14,001–18,000, RMB 18,001–22,000, RMB 22,001–26,000, RMB 26,001–30,000, and more than RMB 30,000). RMB 1.0 is approximately USD 0.8 or EUR 0.7. Participants were also required to indicate if they were a TikTok user (0 = no and 1 = yes), marital status (0 = no and 1 = yes), and educational status (0 = undergraduate and 1 = master student).

### 2.4. Analysis

Before data processing, we removed typos from the data of walking steps (n = 6). Spearman’s rank-order correlation (for two continuous variables), Phi-correlation (for two binary variables), and point-biserial correlation (for a binary variable and a continuous variable) were used to detect the general pattern of associations between the core variables.

The generalized linear model was employed for analysis. Linear and binary logistics were selected as scale responses for processing continuous (while-in-bed screen time and sleep quality) and binary variables (smartphone-related sleep delay and while-in-bed smartphone use), respectively. PA items, including PA duration, frequency, volume, and walking steps, were included as predictors. Meanwhile, participants’ sociodemographic characteristics, including family income, age, gender, education level, and TikTok user, were included as covariates to adjust the models. Variance Inflation Factor (VIF) values smaller than 5.0 were considered evidence of the absence of multicollinearity. Based on this rule, no multicollinearity was observed among the independent variables (VIF < 3.0) [32].

## 3. Results

### 3.1. Participants’ Characteristics

Participant characteristics are shown in Table 1. There were 367 males and 461 females with a mean age of 20.1 years old. The majority of participants were TikTok users (66.4%), undergraduate (97.0%), had family income of RMB 0–6000 (24.5%), performed PA for 20–30 min (23.6%), and used smartphones in bed for 20–30 min (19.8%). Over half of the participants reported bedtime smartphone use (87.8%) and sleep delay (53.1%) yesterday night.

### 3.2. The Associations between Variables of Interest

Table 2 displays the associations between variables of interest. Participants who claimed while-in-bed smartphone use generally showed a greater chance of smartphone-related sleep delay, longer while-in-bed screen time, and worse sleep quality. Participants who claimed smartphone-related sleep delay generally had shorter PA duration, fewer walking steps, longer screen time, and worse sleep quality. While-in-bed screen time was negatively associated with sleep quality.

### 3.3. Total Association between PA Indices and Bedtime Smartphone Use

In binary logistic models, we found a significant association between PA duration and while-in-bed smartphone use (OR = 0.907, *p* = 0.019) (Table 3). The association between PA duration and sleep delay was also statistically significant (OR = 0.932, *p* = 0.014) (Table 3). However, we did not observe any significant association between PA indices and while-in-bed screen time or sleep quality. Notably, we found associations between walking steps and while-in-bed smartphone use and sleep delay, but the estimates were subtle (OR = 1.000).

## 4. Discussion

### 4.1. General Discussion

The presented study aimed to investigate the associations between daytime PA, while-in-bed smartphone use, smartphone-related sleep delay, and sleep quality. We found that greater PA duration was associated lower chance of while-in-bed smartphone use and smartphone-related sleep delay. These results partially support our first and second hypotheses, indicating the positive role of PA in reducing bedtime smartphone use. However, we did not find a distinctive association between PA indices and while-in-bed screen time or sleep quality, not supporting our first and third hypotheses. As far as we know, there are few studies concerning the relationship between PA and bedtime smartphone-related issues. Our previous study found that PA was negatively associated with sleep delay for TikTok use [33]. In the current study, we focused on smartphone-related sleep delay and limited PA to the daytime scenario and observed a similar result. According to the isotemporal substitution model, PA and smartphone use in 24 h may be naturally correlated as one activity occupies time for others [34,35]. However, people are unlikely to perform PA before sleep or in bed. Therefore, our findings under the limited timeframe may collectively imply a potential role of PA in reducing bedtime problematic smartphone use.

Based on published literature, while-in-bed screen use is a common phenomenon that elicits many negative effects, such as increased risks of neck and vision problems [36,37]. Meanwhile, while-in-bed screen use may be associated with sleep delay, a problematic behavior that harms sleep health [21]. This theory is supported by our observed associations between the variables (Table 2). Our results suggest that people who used smartphones in bed tended to delay their sleep and had worse sleep quality. These facts may underline the value of our observed negative associations between PA and while-in-bed smartphone use or sleep delay. Some clues may help explain these findings.

On the one hand, the Internet and electronic devices are recreational methods for many people to cope with stress or negative emotions from school, work, and family [38]. Reports have suggested that stress may be a predictor of problematic smartphone use [38]. Stress is also found to be a reason for smartphone use in Chinese college students [39]. Likewise, some individuals may utilize PA to cope with stress, so stress may also be a predictor of active PA [40]. Therefore, there may be a “competitive inhibition effect” between PA and smartphone use. It is reported that people who are unsatisfied with leisure activities may seek another alternative, such as the Internet, which increases smartphone use [41]. In contrast, people who are more physically active in the daytime may have lower stress levels and, therefore, have less demand for smartphone use in the nighttime. Given that our participants are undergraduates, their PA could basically result from various non-work activities, such as recreational purposes. Therefore, their PA may play a particularly positive role in reducing the chance of nighttime smartphone use.

On the other hand, the hazards of sleep delay are common sense, but most people still cannot simply put down their phones. More than half of our participants claimed sleep delay for smartphone use. These facts indicate that smartphone-related sleep delay can be more than problematic behavior but an addictive behavior [42]. Based on the evidence on exercise rehabilitation for smartphone addiction, both physical exercise and smartphone use may activate similar neurophysiological pathways in the brain [43]. Smartphone use can change the dopaminergic reward circuit and foster habitual behavior [44]. Physical exercise may enhance the synthesis and release of dopamine and stimulates neuroplasticity [45]. Since both forced and voluntary exercise are found to possibly affect reward-related neural plasticity in key reward-based brain structures [43], daily PA may play a similar role in regulating the reward circuit, particularly for those recreational physical activities. In addition, daily PA may increase inhibitory control [46], which may reduce addictive behaviors, including addictive smartphone use [47].

Although we found distinct associations between PA duration and smartphone-related sleep delay, the association between PA duration and while-in-bed screen time was not statistically significant. These results appear contradictory. One possible reason is that participants who performed longer PA might have lied in bed earlier, so they might be able to keep their while-in-bed screen time without delaying sleep. However, this cannot be confirmed based on our results and may remain a further research topic. We did not find a distinct association between sleep quality and PA indices. Insufficient sleep may be a critical reason for poor sleep quality in young people [48]. Our investigation was on the weekend, so most of our participants could get up late and have enough sleep, which may be a reason for the null results.

### 4.2. Strengths and Limitations

Many studies have documented the negative associations between PA and smartphone use. However, the causal relationship between the two variables cannot be confirmed due to the cross-sectional design [49]. Some scholars have assumed smartphone use limits PA and vice versa [33,50]. The presented study is also based on a cross-sectional investigation, but the investigated variables have a time sequence. Although the causal relationship still cannot be fully confirmed due to numerous factors involved and the periodicity of smartphone use and PA, our findings may somewhat offer evidence to support the positive effects of PA on nighttime smartphone use. Thus, our findings may help us understand the reciprocal relationship between PA and smartphone use.

There are some limitations to this study. Due to our limited experimental conditions, we only use self-reports to capture data on sleep-related variables, so informant bias must have existed. Future investigation or controlled trials are recommended to employ objective measures. For example, actigraphy is an ideal method for inferring sleep/wake patterns as it can monitor activity in natural settings [51]. This method is widely used to study sleep disorders and is expected to help understand smartphone-induced sleep issues in young people [52].

We only deployed qualitative questions to investigate sleep delay, while the exact duration of sleep delay was not involved. Further work may need to pay more attention to the sleep delay duration. For example, measuring their expected and actual sleep time may help assess the duration of sleep delay. Moreover, future work should also consider the dose-response relationship between sleep delay duration and sleep quality.

Our study was conducted on the weekend. Weekend behaviors are theoretically distinct from weekday behaviors, which may affect people’s demand for recreational activities and limit us from generalizing the findings.

We only controlled the sociodemographic variables in the regression, and factors concerning other daytime habits were not investigated, particularly for those routine activities on weekdays. These factors may elicit plausible residual effects on sleep quality and therefore need to be considered in future work.

We mainly used subjective measures due to our limited experimental conditions. We tried to capture data on PA based on smartphone sensors, but the results were not highly relevant to results obtained in self-reports. This may be because smartphones can be held in hands, pockets, and bags. Students would not carry their smartphones while doing sports or exercise, such as gym training or playing basketball. Hence, many activities might not have been recorded by smartphones. Future research may use specialized devices to capture data on PA, screen time, and sleep quality.

## 5. Conclusions

The current study aimed to investigate the effects of daily PA on nighttime smartphone use. We found that self-reported PA duration was associated with lower chances of while-in-bed smartphone use and smartphone-related sleep delay. However, we found no effects of PA on while-in-bed screen time and sleep quality. These findings may support the role of daily PA in reducing problematic smartphone use. Due to our research design and limited research methods, we cannot offer a strong asseveration, and all the clues must be concluded and generalized to the public with caution. For this reason, we call for additional studies to re-examine the identified associations with advanced methods. Additionally, since this is a preliminary study that only offers theoretical indications, we call for controlled trials with designed programs to offer practical recommendations for public health promotion.

## Figures and Tables

**Figure 1 ijerph-19-09693-f001:**
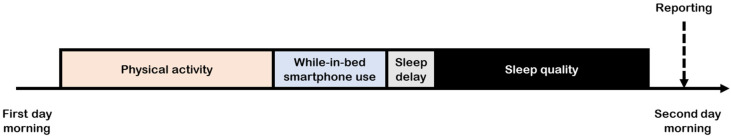
The time sequence of the investigated variables.

**Table 1 ijerph-19-09693-t001:** Participants’ characteristics.

Variable	Category	Mean (SD)	Percentage (*N*)
Gender	Male	-	44.3% (367)
	Female	-	55.7% (461)
Marital status	Unmarried		95.5% (791)
	Married		4.5% (37)
Age (year)	-	20.11 (2.3)	-
Monthly income (CNY)	0-6000	-	24.5% (203)
	6001–10,000	-	18.6% (154)
	10,001–14,000	-	11.1% (92)
	14,001–18,000	-	7.1% (59)
	18,001–22,000	-	4.1% (34)
	22,001–26,000	-	2.5% (21)
	26,001–30,000	-	6.9% (57)
	>30,000	-	24.5% (203)
Educational status	Undergraduate	-	97.0% (803)
	Master student	-	3.0% (25)
TikTok users	User	-	69.4% (575)
	Non-user	-	30.6% (253)
PA duration	0–10 min	-	9.8% (81)
	10–20 min		15.9% (132)
	20–30 min	-	23.6% (195)
	30–40 min	-	18.4% (152)
	40–50 min	-	6.8% (56)
	50–60 min	-	9.7% (80)
	60–70 min	-	5.1% (42)
	70–80 min	-	1.2% (10)
	80–90 min	-	1.0% (8)
	>90 min	-	8.7 (72)
PA intensity (point)	-	5.4 (2.4)	-
PA volume (point)	-	23.9 (20.4)	-
Walking steps (steps)		9334.8 (5846.3)	-
While-in-bed screen time	0–10 min	-	12.2% (101)
	10–20 min		11.7% (97)
	20–30 min	-	19.8% (164)
	30–40 min	-	15.1% (125)
	40–50 min	-	8.5% (70)
	50–60 min	-	12.3% (102)
	60–70 min	-	5.4% (45)
	70–80 min	-	1.9% (16)
	80–90 min	-	2.4% (20)
	>90 min	-	10.6% (88)
While-in-bed smartphone use	Yes	-	87.8% (727)
	No	-	12.2% (101)
Smartphone-related sleep delay	Yes	-	53.1% (440)
	No	-	46.9% (388)
Sleep quality (point)	-	7.6 (2.4)	-

Note: PA, physical activity.

**Table 2 ijerph-19-09693-t002:** Association between core variables.

	1	2	3	4	5	6	7	8
1. PA duration	1.000							
2. PA intensity	0.254 **	1.000						
3. PA volume	0.824 **	0.708 **	1.000					
4. Walking step	0.321 **	0.058	0.245 **	1.000				
5. Phone use	−0.063	0.027	−0.029	−0.087	1			
6. Sleep delay	−0.085 *	0.010	−0.059	−0.097 *	0.286 **	1		
7. Screen time	−0.033	−0.010	−0.024	−0.061	0.356 **	0.360 **	1.000	
8. Sleep quality	0.068	0.017	0.040	0.033	−0.143 **	−0.229 **	−0.199 **	1.000

Note: PA, physical activity. Phone use, while-in-bed smartphone use; Sleep delay, smartphone-related sleep delay; Screen time, while-in-bed screen time. *, *p* < 0.05; **, *p* < 0.01.

**Table 3 ijerph-19-09693-t003:** The associations between PA items and variables of interests.

Variable	PA Indices	OR (95%CI)	*p*
While-in-bed smartphone use	PA duration	**0.907 (0.836, 0.984)**	**0.019**
	PA intensity	1.044 (0.954, 1.143)	0.348
	PA volume	0.994 (0.984, 1.004)	0.240
	Walking steps	1.000 (1.000, 1.000)	0.045
Smartphone-related sleep delay	PA duration	**0.932 (0.881, 0.986)**	**0.014**
	PA intensity	1.012 (0.954, 1.073)	0.697
	PA volume	0.994 (0.987, 1.001)	0.994
	Walking steps	1.000 (1.000, 1.000)	0.039
	PA indices	B (95%CI)	*p*
While-in-bed screen time	PA duration	−0.043 (−0.117, 0.030)	0.250
	PA intensity	−0.045 (−0.123, 0.032)	0.253
	PA volume	−0.001 (−0.010, 0.008)	0.813
	Walking steps	−0.000 (−0.000, 0.000)	0.057
Sleep quality	PA duration	0.036 (−0.028, 0.101)	0.271
	PA intensity	0.020 (−0.049, 0.088)	0.574
	PA volume	0.005 (−0.003, 0.013)	0.186
	Walking steps	0.000 (−0.000, 0.000)	0.821

Note: PA, physical activity. Coefficients are unstandardized linear regression coefficients (B) or odds ratios (OR) with 95% confidence intervals (CI) and significance (*p*-values) reported and shown in bold text.

## Data Availability

The data is available upon request from the corresponding author.

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
