# Peer review of "The Associations between Daytime Physical Activity, While-in-Bed Smartphone Use, Sleep Delay, and Sleep Quality: A 24-h Investigation among Chinese College Students"

_ijerph, 2022, doi:10.3390/ijerph19159693_

Round 1

Reviewer 1 Report

I find the article is interesting. However, I suggest the authors in the section of discussion to discuss in more details if there is contradictions with previous studies in the same topic. I suggest for the authors to put some medical flavors (medical information) related to the disorders in sleep related to smart phone use, the possible mechanisms and melatonin relation. 

Reviewer 2 Report

This is a well-written paper on a very interesting topic, aiming to investigate the relation between physical activity and sleep among a target population of students.

Due to its methodology several limitations exists, but I found then clear enough mentioned in the corresponding section. I would only suggest to include the role of actigraphy as a objective measure of sleep and wake, for future investigations.

Because of the design of the investigation, not too strong asseverations could be done. This should be highlighted and this investigation may only serve as a starting point for others.

Please, review the spelling on “Smartphone use and health in-91 vestigation” and sent the recruiting massage to 50 classes in two colleges” in page 2.

Reviewer 3 Report

Dear Authors,

It is an interesting article; I add some suggestions:

Physical activity is used many times; I suggest to use the abbreviation PA to reduce the long of the paper.

How was measure the sleep quality? It will be nice to include a line in the in the abstract.

More details should be presented about the conclusion of this study, how long need to be the duration of PA? what happen if the PA finish early during the day, at 16, 17, 18? It means that they are not going to use the smartphone or any other device while-in be? I think the conclusion cannot be supported only by the presented data, so it should be indicated

Keywords. Do not use words already presented in the title of the article

To delete the abbreviation GLM, it is never used in the document

All abbreviations must be explained, also in the tables, PA, OR….

To delete unnecessary decimals, for readers is better 20.1 than 20.11, 44.3% than 44.32%, etc…

Best Regards
